# Agonists of melatonin receptors strongly promote the functional recovery from the neuroparalysis induced by neurotoxic snakes

**Giorgia D'Este[1‡], Federico Fabris[1‡], Marco Stazi[1¤], Chiara Baggio[1], Morena Simonato[2], Aram Megighian[1,3], Michela Rigoni[1,4], Samuele Negro[1,5]\*, Cesare Montecucco[1,2]\***

1 Department of Biomedical Sciences, University of Padova, Padova, Italy, 2 CNR Institute of Neuroscience, Padova, Italy, 3 Padua Neuroscience Center, University of Padova, Padova, Italy, 4 Myology Center (CIR-Myo), University of Padova, Padova, Italy, 5 U.O.C. Clinica Neurologica, Azienda Ospedale-Università Padova, Padova, Italy

¤ Current address: The Francis Crick Institute, London, United Kingdom
‡ These authors share first authorship on this work.
\* samuele.negro1987@gmail.com (SN); cesare.montecucco@gmail.com (CM)

**Data Availability Statement:** All relevant data are within the manuscript and its Supporting information files.

## Abstract

Snake envenoming is a major, but neglected, tropical disease. Among venomous snakes, those inducing neurotoxicity such as kraits (*Bungarus* genus) cause a potentially lethal peripheral neuroparalysis with respiratory deficit in a large number of people each year. In order to prevent the development of a deadly respiratory paralysis, hospitalization with pulmonary ventilation and use of antivenoms are the primary therapies currently employed. However, hospitals are frequently out of reach for envenomated patients and there is a general consensus that additional, non-expensive treatments, deliverable even long after the snake bite, are needed. Traumatic or toxic degenerations of peripheral motor neurons cause a neuroparalysis that activates a pro-regenerative intercellular signaling program taking place at the neuromuscular junction (NMJ). We recently reported that the intercellular signaling axis melatonin-melatonin receptor 1 (MT1) plays a major role in the recovery of function of the NMJs after degeneration of motor axon terminals caused by massive $Ca^{2+}$ influx. Here we show that the small chemical MT1 agonists: Ramelteon and Agomelatine, already licensed for the treatment of insomnia and depression, respectively, are strong promoters of the neuroregeneration after paralysis induced by krait venoms in mice, which is also $Ca^{2+}$ mediated. The venom from a *Bungarus* species representative of the large class of neurotoxic snakes (including taipans, coral snakes, some Alpine vipers in addition to other kraits) was chosen. The functional recovery of the NMJ was demonstrated using electrophysiological, imaging and lung ventilation detection methods. According to the present results, we propose that Ramelteon and Agomelatine should be tested in human patients bitten by neurotoxic snakes acting presynaptically to promote their recovery of health. Noticeably, these drugs are commercially available, safe, non-expensive, have a long bench life and can be administered long after a snakebite even in places far away from health facilities.

**Funding:** This work was supported by the project "RIPANE" of the Italian Ministry of Defense (MONT_COMM18_03) to CM and by funding from the University of Padova (DOR to MR), The funders had no role in study design, data collection and analysis, decision to publish, or preparation of the manuscript.

**Competing interests:** The authors have declared that no competing interests exist.

## Author summary

Snakebite envenomings cause important tropical human diseases that often include a lethal muscle paralysis. Current treatments consist in hospitalization and antivenoms, which are not always quickly accessible to victims. In fact, these snakebites take place mainly in rural and low-income countries.

In this work, researchers discovered, in mice, a novel function of melatonin and of its type 1 receptor in promoting functional recovery after snake-induced peripheral neuroparalysis with nerve terminal degeneration. In particular, researchers found that drugs approved for the treatment of insomnia (Ramelteon) and depression (Agomelatine), activate melatonin receptor and promote the functional recovery after a krait venom induced paralysis.

These drugs are on sell in pharmacies, are safe and stable, and are ready to be tried for promoting the recovery from peripheral neuroparalysis in human victims bitten by neurotoxic snakes, even without hospitalization.

## Introduction

Venomous snakes were estimated to strike over two and a half million people every year, causing more than 100,000 deaths mainly in tropical and sub-tropical areas of the world. About a fourfold larger number of envenomed patients develop permanent physical and psychological sequelae, with high social costs [1–5]. Envenomation exhibits heterogeneity in its clinical manifestations and is believed by experts to be under reported, also because it mainly occurs in parts of the world characterized by a limited accessibility to hospitals or other healthcare facilities [1–7].

A large number of the life-threatening snakebites, including those by snakes belonging to the families Elapidae and Viperidae, cause an acute peripheral neuroparalysis with impaired lung ventilation [8,9]. Snakes of the *Bungarus* genus include at least nineteen species widely distributed in South and East Asia [10–21]. These snakes are termed kraits and their venoms induce a descending flaccid paralysis with respiratory failure, general weakness, ptosis, diplopia, difficult swallowing, dysarthria, abdominal pain and autonomic dysfunctions closely resembling botulism and the Guillain-Barrè autoimmune syndromes [9–32]. A rapid and large reduction in function of the neuromuscular junction (NMJ) occurs within a few hours from snakebite [9]. If respiration is not assisted, death may occur, mainly by respiratory failure. Mechanically ventilated patients may survive because the *Bungarus* neurotoxins do not kill motor neurons, rather they induce a rapid degeneration limited to the axon terminals which is followed by a regeneration that takes about one week in mice and a variable number of weeks in humans. This depends on many factors including patient's health, amount of injected venom, snake species, bite site and others [9–32]. Accordingly, envenoming by kraits in the absence of the early injection of an antivenom snake species specific, may require prolonged hospitalization with lung ventilation, but, eventually, the patients recover. However, respiratory muscle weakness in these patients admitted to intensive Care Units due to prolonged mechanical ventilation contributes to difficult ventilator weaning and poor long-term outcomes [33]. Moreover, long hospitalization with mechanical support of respiration may not be available in low-income countries. In any case, hospitalization in intensive care units increases medical costs and the risk of in-hospital complications such as nosocomial infections and pulmonary embolism. This situation calls for the identification of small molecules acting as therapeutics that do not strictly depend on the biting snake species and that can be used in low-income settings where intensive care units may not be available [1,2].

The peripheral neurotoxicity of kraits, taipans, coral snakes and some vipers is due to the action of pre-synaptic and post-synaptic neurotoxins [8,34]. Snake phospholipase A2 toxins (β-bungarotoxins, β-BTX and related neurotoxins) acting presynaptically are expressed by all the above-mentioned classes of snakes. The other major type of neurotoxin present in the venom of these snakes is a curare-mimetic neurotoxin (α-bungarotoxin, α-BTX) that binds the post-synaptic acetylcholine receptor thus preventing muscle stimulation [8,34].

The β-BTXs cleave the ester bond of the phospholipids in the sn-2 site of the glycerol moiety of phospholipids with a progressive hydrolysis of membrane phospholipids [35–41]. The change in cell membranes composition and structure consequent to the large production of lysophospholipids and fatty acids (LP and FA) is responsible for mitochondrial alterations with loss of function and a massive discharge of synaptic vesicles not followed by retrieval [35,36]. In fact, when the concentration of LP/FA is sufficiently high, transient lipidic channels form in the membrane, allowing $Ca^{2+}$ flow across the membrane along its concentration gradient [41–43]. High cytosolic [$Ca^{2+}$] also causes the opening of the mitochondrial transition pore [44] and the activation of cytosolic hydrolases with the consequent degeneration of axon terminals [45] that is completed within a few hours.

Remarkably, this degeneration is followed by the regrowth of motor axon terminals and reformation of functional NMJs. Given the importance of NMJ functionality for survival, peripheral neuroregeneration has been conserved through evolution. Its progression can be monitored by imaging the NMJ using pre- and post-synaptic markers and by electrophysiology; in the lung it can be followed with electromechanical methods.

A transcriptional analysis of the intoxicated NMJ led us to discover two intercellular signalling axis that are essential for neuroregeneration. The first one consists of the CXCR4 receptor, re-expressed at the axon stump upon nerve terminal degeneration, and its ligand, the chemokine CXCL12α, which is released by activated perisynaptic Schwann cells (PSCs) [46]. More recently, we demonstrated that melatonin binding to MT1 receptors, expressed by PSCs upon injury, is also involved in neuroregeneration [47]. In experiments in human patients affected by different neuropathological conditions, melatonin has been used in doses as high as 300 mg/day [48–51]. Therefore, we have decided to test the effect of two MT1 agonists, that are effective at much lower doses and have been already approved for human therapy for the treatment of insomnia and depression, respectively [52,53], Ramelteon (commercial name Rozerem) and Agomelatine (commercial names: Valdoxan and Thymanax) (S1 Fig). This also implies that they are safe and commercially available.

We report here on the very positive effect of Ramelteon and Agomelatine on the recovery of NMJ function after degeneration of motor axon terminals caused by the venom of *Bungarus caeruleus*, a venom containing β-BTX. These drugs cause a relevant recovery of the respiratory function with respect to mice treated with vehicle already after 24 hours from venom injection. This venom induces a neuroparalysis which is very similar to those caused by other kraits, by taipans, by coral snakes and by some vipers and, therefore, the results reported here are very likely to be valid also for other neurotoxic snakes. Based on these results, we suggest testing the effect of Ramelteon and Agomelatine on the recovery of normal physiology in neuroparalysed patients bitten by neurotoxic snakes.

## Materials and methods

### Ethical statement and experimental model

Procedures carried out in Italy were approved by the ethical committee and by the animal welfare coordinator of the OPBA from the University of Padua. All procedures are specified in the projects approved by the Italian Ministry of Health, Ufficio VI (authorisation numbers: 359/

2015 PR and were conducted in accordance with National laws and policies (D.L. n. 26, March 14, 2014), following the guidelines established by the European Community Council Directive (2010/63/EU) for the care and use of animals for scientific purposes. Animals were handled by specialised personnel under the control of inspectors from the Veterinary Service of the Local Sanitary Service (ASL 16-Padua), who are the local officers of the Ministry of Health. All procedures were performed under general anesthesia and analgesia and tissue samples were collected from mice sacrificed under deep anesthesia. Mice expressing cytosolic green fluorescent protein (GFP) in Schwann cells (SCs) under the plp promoter, kindly provided by Dr. W.B. Macklin (Aurora, Colorado), were used in some immunofluorescence experiments. Electrophysiological recordings, immunofluorescence, and lung ventilation experiments were conducted using 25–30 gr CD1 mice bred in the animal house at the University of Padua. Mice were maintained under a 12 hours light/dark cycle in the animal facility of the Department and kept under constant temperature. Water and food were available ad libitum, and mice were fed with regular chow. Paralysis induced by *Bungarus caeruleus* did not impair food or water intake.

## Venom and reagents

*Bungarus caeruleus* venom was supplied by Latoxan (France) and the lyophilized samples were kept at –20˚C and reconstituted in phosphate-buffered saline solution (PBS) prior to use. The following primary antibodies were employed: anti-VAMP1 (1:200) [54], anti-Neurofilament (Abcam, cat. Ab 4680, 1:800), mAb-A06 (MT1) provided by Prof. M. Solimena [55], α-bungarotoxin Alexa555-conjugated (α-BTX555) (cat. B35451, 1:200) and secondary antibodies Alexa-conjugated (1:200) were from Life Technologies. Agomelatine (cat. A1362), Ramelteon (cat. SML2262), and melatonin (cat. M5250) were purchased from Sigma-Aldrich. Unless otherwise stated, all other reagents were from Sigma.

## Evoked Junctional Potentials (EJPs) recordings

Electrophysiological measurements were performed in 6–8 week old CD1 mice weighting 25–30 gr. Upon isoflurane anaesthetization, mice were locally injected in the hind limb with 36 μg/Kg of *B. caeruleus* venom diluted in 15 μl PBS containing 0.2% gelatin. Mice were daily locally injected with Ramelteon (29 μg/Kg, diluted in 20μl PBS containing 0.2% gelatin) or vehicle or Agomelatine (0.36 mg/Kg) in the soleus muscle. The first injection of agonists was performed after 4 hours from *B. caeruleus* venom injection. For Ramelteon we used a dose that corresponds to one fourth of that suggested for long-term use in humans (8 mg/day, taking 70 Kg as average weight), while for Agomelatine we employed the same dose used in humans (25 mg/day). Ninety-six hours later, soleus muscles were quickly excised and pinned on the bottom of a Sylgard-coated petri dish (Sylgard 184, Down Corning USA). Recordings were performed in oxygenated (95% $O_2$, 5% $CO_2$) Krebs-Ringer solution using intracellular glass microelectrodes (1.5 mm outer diameter, 1.0 mm inner diameter, 15–20 MΩ tip resistance; GB150TF, Science Products GmbH Germany), filled with a 1:2 solution of 3 M KCl and 3 M CH3COOK. Evoked neurotransmitter release was recorded in current-clamp mode, and resting membrane potential was adjusted with current injection to -70 mV. EJPs were elicited by supramaximal nerve stimulation at 0.5 Hz using a suction glass microelectrode (GB150TF, Science Products GmbH Germany) connected to a S88 stimulator (Grass, USA). Muscle fiber contraction during intracellular recordings was blocked by adding 1 μM μ-Conotoxin GIIIB (Alomone Lab, Israel). Intracellularly recorded signals were amplified with an intracellular amplifier (BA-01X, NPI, Germany). Using a digital A/C interface (NI PCI-6221, National Instruments, USA), amplified signals were converted to digital format and fed to a PC for

offline analysis and online visualization using the relevant software (WinEDR, Strathclyde University; pClamp, Axon, USA). Stored data were analyzed off-line using the software pClamp (Axon, USA). At the end of the experiment, the muscles were processed for immunofluorescence.

## Compound muscle action potential (CMAP) recordings

Upon isoflurane anaesthetization, mice were injected with 36 μg/Kg *B. caeruleus* venom in the hind limb. Mice where daily i.p. injected, without anesthesia, for 4 days with the doses of Ramelteon or Agomelatine given above. The first injection of agonists was performed after 4 hours from *B. caeruleus* venom injection. The analysis was performed 4 days after intoxication. The sciatic nerve was exposed at the trochanter level after general anesthesia without injuring the gluteal musculature. A tiny piece of parafilm was placed beneath the nerve and kept moist with a drop of saline (0.9% NaCl in deionized water). A pair of stimulating needle electrodes (Grass, USA) were advanced until they gently touched the exposed sciatic nerve using a mechanical micromanipulator (MM33, FST, Germany). A pair of needle electrodes for electro-myography (Grass, USA) were used for electromyographic recording of gastrocnemius muscle fibers activity. The recording needle electrode was placed midway into the gastrocnemius muscle, while the indifferent needle electrode was inserted in the distal tendon of the muscle.

Compound muscle action potentials (CMAPs) were recorded following supramaximal stimulation of the sciatic nerve at 0.5 Hz (0.4 ms stimulus duration) using a stimulator (S88, Grass, USA) via a stimulus isolation unit (SIU5, Grass, USA) in a capacitive coupling mode. An extracellular amplifier (NPI, Germany) was used to amplify the recorded signals. A digital A/C interface (see above, National Instruments, USA) was then used to digitize the signals, and the signals were sent to a PC for offline and online analysis using appropriate softwares (pClamp, Axon, USA; WinEDR, Strathclyde University. Stored data were analyzed offline using pClamp software (Axon, USA).

## Inferred ventilation index (IVI) recordings

The respiratory impairment, which can be fatal, is one of the main signs of krait envenomation. For this reason, we evaluate how Ramelteon and Agomelatine affect the ability of envenomed mice to recover the respiratory function. An indirect, highly sensitive, test was used to measure this physiological parameter. A probe connected to a pressure sensor was placed inside the oesophagus at the mediastinum level in anesthetized mice. Asymmetric peaks that match the frequency of lung ventilation events characterize the signal pattern we recorded. The peak area may be taken as an estimate of the volume of air inspired, which is directly related to pressure variations resulting from the activity of muscles involved in breathing. Following general anesthesia, animals were left 10 minutes in their cages to relax. For each mouse, recordings were performed at t = 0 and 24, 96 and 168 hours after intoxication with 36 μg/Kg *B. caeruleus* venom injected in chest muscles (diluted in 40 μl PBS containing 0.2% gelatin). Mice were i.p. injected daily for 4 days with the doses of Ramelteon or Agomelatine reported above. A 20 ga x 38 mm plastic feeding tube (Instech Laboratories, Inc.) attached to a pressure sensor (Honeywell, 142PC01D) was carefully positioned in the mice oral cavity and placed in the lower third of the esophagus at the level of the mediastinum. Animals were laid on their left side on a pre-warmed surface to record their lung pressure variations, which were used to infer animal ventilation by recording signals that were amplified and digitized with WinEDR V3.4.6 software (Strathclyde University, Scotland). Stored data were analyzed using Clampfit software (Axon, USA). We recorded 2 minutes of ventilation for each animal to ensure trace stability, and then 20 seconds of each trace were analysed. Inferred ventilation index (IVI) was

then calculated as the product of the average area of the peaks (mV x ms) and the number of peaks within 20 sec.

## Immunofluorescence

Anesthetized mice were locally injected with a *B. caeruleus* venom dose of 36 μg/Kg in the hind limb or in intercostal muscles w/wo Ramelteon or Agomelatine. Soleus, gastrocnemius and different respiratory muscles were dissected at different time points, fixed in 4% paraformaldehyde in PBS for 30 min at room temperature, and quenched in PBS containing 0.24% $NH_4Cl$. After 2h of permeabilization and saturation in blocking solution (15% goat serum, 2% BSA, 0.25% gelatin, 0.20% glycine, 0.5% Triton X-100 in PBS), samples were incubated with primary antibodies for 72 h in blocking solution at 4˚C. Muscles were then washed three times in PBS and incubated with secondary antibodies diluted in PBS+0.5% Triton X-100 for 2 h at room temperature. After 3 additional washes in PBS for 10 minutes, samples were mounted using Dako fluorescence mounting medium (Agilent Technologies, cat S3023). Images were collected with a Zeiss LSM 900 Confocal microscope equipped with a 40× HCX PL APO NA 1.4 oil immersion objective. To reduce crosstalk, the laser excitation line, power intensity, and emission range were chosen based on the specific fluorophore of each sample.

## Quantification and statistical analysis

Sample sizes were determined based on data collected in published studies. We used at least N = 4 mice/group for electrophysiological analysis. We ensured a blind conduct of experiments and imaging analysis. Data were displayed as histograms and expressed as means ± SD. GraphPad Prism software was used for statistical analyses. Statistical significance was evaluated using unpaired Student's t-test or by one-way analysis of variance (ANOVA) with Tukey's post-test when more than 2 experimental conditions were compared. Data were considered statistically different when * $p<0.05$, ** $p<0.01$, *** $p<0.001$ and **** $p<0.0001$.

## Results

### Melatonin receptor type 1 is expressed by Schwann cells following motor axon terminals degeneration induced by *B. caeruleus* snake venom

The venom of the common krait (*Bungarus caeruleus*) was chosen given its wide territorial distribution, and as the *Bungarus* species responsible for a large numbers of cases of severe neurotoxic envenoming each year. In addition, *Bungarus caeruleus* envenomation is representative of those caused by other *Bungarus* species [6,8–21].

We recently showed that melatonin receptors type 1 (MT1) associated to PSCs are involved in the recovery of motor axon terminals after degeneration induced by calcium overload caused by the pore forming toxin α-latrotoxin [47]. To test whether this occurs also in the neurodegeneration caused by a *Bungarus* venom, we collected the muscle samples at different time points of the regeneration period (Fig 1A) and performed an immunofluorescence analysis whose results are shown in Fig 1B. The venom of *B. caeruleus* causes a structural degeneration of the motor axon terminal with loss of the neurofilament (NF) staining, whilst leaving intact the convoluted post-synaptic structure of the murine NMJ, as seen by α-BTX staining. NF staining almost disappeared after 24 h from i.m. venom injection, indicating a complete degeneration of the motor axon terminal. Remarkably, NF staining recovers with time and returns almost to control by seven days in envenomed mice. In parallel, we observed the expression of MT1 on PSCs whose signal persists for several days untill it declines by seventh day from injection.

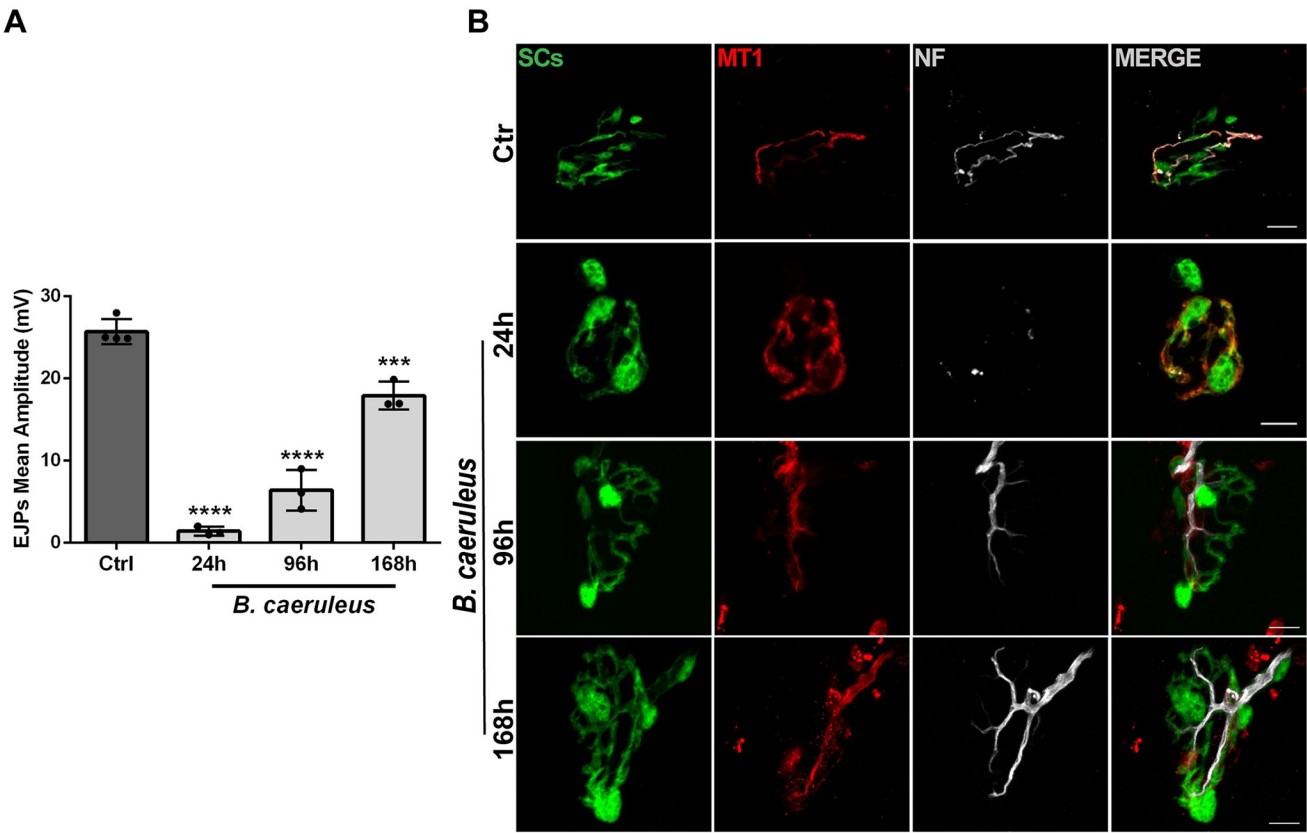

**Fig 1.** *Bungarus caeruleus* **venom causes an acute and reversible degeneration of motor axon terminals at the soleus NMJs and induces the expression of melatonin receptor type 1 (MT1) in Perisynaptic Schwann cells.** A) *B. caeruleus* venom was injected intramuscularly (i.m.) in the hind limb of anesthetized mice, and regeneration was followed via electrophysiological recordings of the Evoked Junction Potentials (EJPs) at 24h, 96h and 168h after intoxication. Each bar represents mean ± SD. N = 4 animals (Ctrl), N = 3 animals (24h, 96h, 168h), 15 fibres/animal. ****$p<0.0001$ ANOVA followed by post hoc Tukey test. B) Representative immunostaining of soleus NMJs from GFP-expressing mice. Soleus muscles were collected after intoxication with *B. caeruleus* at different time points (0, 24 h, 48 h, 96 h, 168 h), and stained with specific antibodies. Motor axon terminals are identified by Neurofilament heavy chain (NF-H, *white*), PSCs by endogenous GFP (*green*) and MT1 (*red*) by a specific monoclonal antibody [58]. Scale bars: 10 μm.

These findings provide the cellular basis for the testing of melatonin and MT1 agonists on the regeneration of NMJ in envenomated animals. The finding of any positive activity of melatonin and agonists targeting MT1 receptors would indicate that these drugs display their regenerative power by acting on PSCs, whose action is known to be essential for the functional and anatomical regeneration of motor axon terminals after degeneration [56–59].

## Agonists of melatonin receptor type 1 strongly promote the functional regeneration of the motor axon terminals after degeneration induced by *B. caeruleus* venom

To estimate the effect of agonists of MT1 receptor on the recovery of the neuromuscular function after nerve degeneration induced by snake venom injection, we recorded the evoked junctional potential (EJP) (Fig 2A, 2B and 2E) and the compound muscle action potential (CMAP) (Fig 2C, 2D and 2F). Melatonin administration (25 mg/Kg) was capable of promoting the recovery of the end plate potential of single fibers of the soleus muscle injected intramuscularly with *B. caeruleus* venom (Fig 2B, white bar). However, as Melatonin was reported to be effective in neurodegenerative disorders in humans at much higher dose, we look for alternative

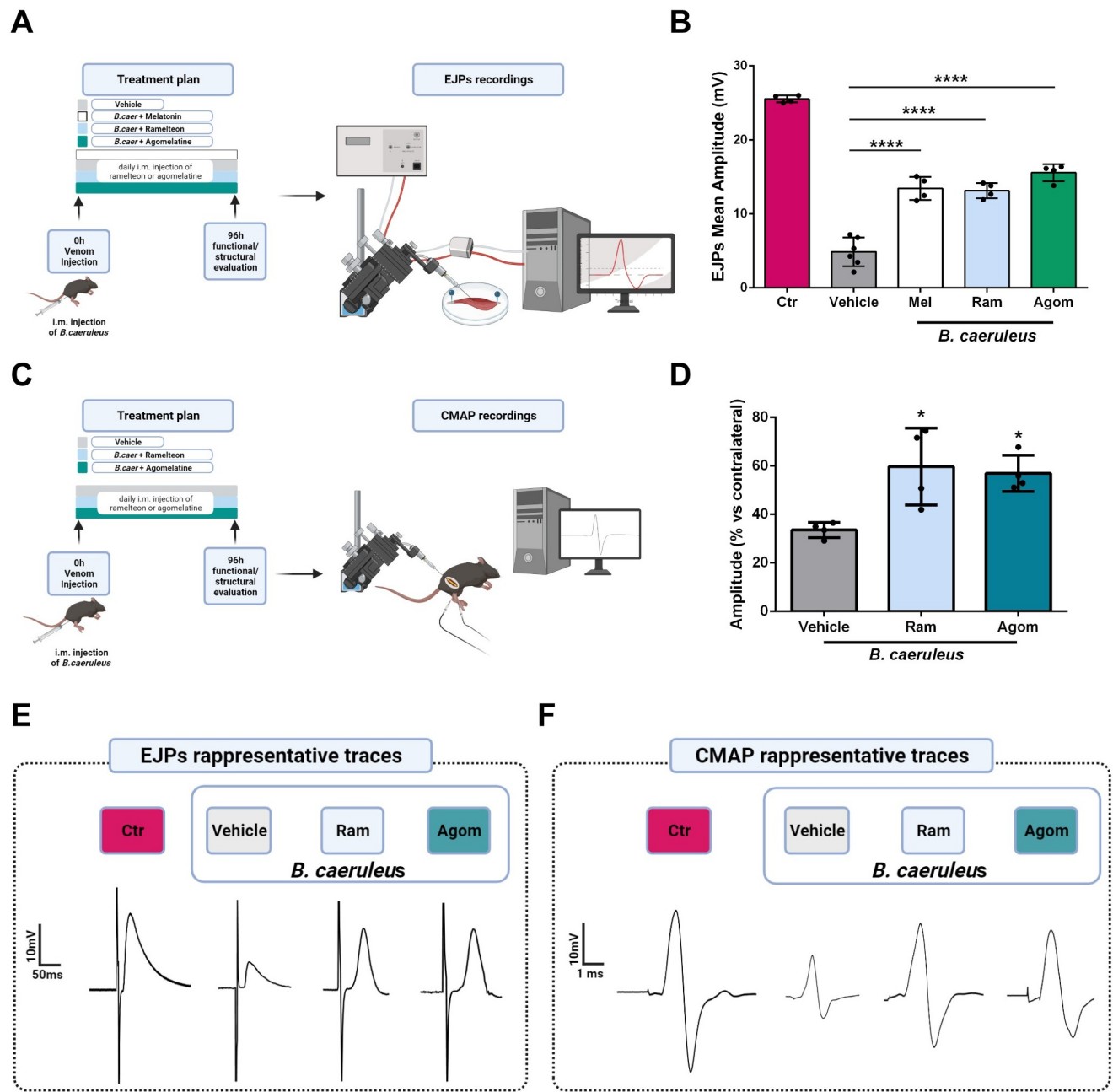

**Fig 2. The MT1 agonists Ramelteon and Agomelatine strongly promote recovery of function of the NMJ after injection of *B. caeruleus* venom.** A) Schematic representation of the technique employed to measure the Evoked Junctional Potentials (EJPs), Created with BioRender.com. B) EJPs of soleus muscles 96 hours post injection of *B. caeruleus* venom in the hind limb of mice treated daily with melatonin (Mel), Ramelteon (Ram), or Agomelatine (Agom). Each bar represents the mean of the EJP amplitude ± SD, at least N = 4, number of analyzed fibers: 15/mice, One way ANOVA ****p <0,0001. (C) Schematic representation of the technique employed for CMAP measurements. D) CMAP amplitudes recorded in gastrocnemius muscles 96 h post injection of *B. caeruleus* venom in the absence or presence of Ramelteon and Agomelatine. N = 4, One-way ANOVA *p< 0.05. E-F), Representative EJP (E) and CMAP (F) traces of control and treated muscles w/o Ramelteon or Agomelatine.

molecules acting on MT1 receptors that are active at safe doses and commercially available for use in humans, and thus we opted for two small and stable molecules: Ramelteon and Agomelatine. Ramelteon has been approved by FDA for the long-term treatment of insomnia at the dose of 8 mg/day [52] whilst Agomelatine has been approved by EMA for the long-term

treatment of depression at 25 mg/day [53]. In mice we used Ramelteon at a dose of one fourth of that suggested for humans since preliminary experiments with different doses indicated a relevant effect even at the lowest dose tests. Agomelatine was administered at the same dose as the one suggested for humans. The drugs were injected in the soleus or gastrocnemius muscles daily, depending on the experimental plan (Fig 2A and 2C). Recording the Evoked Junctional Potential (EJP) allows one to estimate neurotransmitter release at single NMJ providing a quantitative assay of the nerve terminal functionality. This parameter was determined after 96 hours from injection, which is an intermediate time point of the recovery period of the soleus NMJs (Fig 1A) [60].

Fig 2B shows that the daily treatments for 4 days of both Ramelteon and Agomelatine, used at doses well tolerated by humans for long periods, are very effective in promoting the recovery of the synaptic function within the NMJ at the single muscle fiber level. EJPs traces from muscle fibers of mice treated with Ramelteon and Agomelatine show higher amplitude than those treated with vehicle only, further supportive of enhanced recovery (Fig 2E).

An additional relevant electrophysiological assay used in the present study is the compound muscle action potential (CMAP) (Fig 2C), which evaluates the recovery of function of the entire muscle, and therefore is more adherent to the clinical observation [61]. Fig 3D shows that the extent of recovery of CMAP 96 h from the injection of *B. caeruleus* venom is greatly increased by the daily treatment with Ramelteon or Agomelatine. This result is further confirmed by the greater amplitude of CMAP traces of muscles treated with Ramelteon and Agomelatine compared to those with vehicle alone (Fig 2F).

In the same muscle subjected to EJP and CMAP recordings, the therapeutic effects of Ramelteon and Agomelatine were also evaluated by an immunofluorescence analysis of NMJ structure using as post-synaptic marker the acetylcholine receptor, stained with fluorescently labelled α-BTX, and the presynaptic marker, VAMP-1 (Fig 3A). VAMP-1 signal and distribution, clearly shows that both MT1 agonists speed up the recovery of presynapse with respect to mock controls. The extent of recovery was estimated by calculating the percentage of denervated, partially innervated, and innervated NMJs, as well as the NMJ occupancy, which characterizes the overlap between the presynaptic and postsynaptic morphology [62]. Results are reported in panels B, C, of Fig 3 and parallel the electrophysiological findings.

The same morphological analysis was performed in the gastrocnemius muscles used for the measurement of CMAP. The immunofluorescence staining is shown in Fig 3D; again, the ratio of pre- and post-synaptic staining was determined and a remarkable parallelism with the electrophysiological measures (Fig 3E and 3F) was found.

Taken together, the data presented in Figs 2 and 3 documents that Ramelteon and Agomelatine strong stimulate neuroregeneration of paralyzed motor axon terminals completely paralyzed by the action of the venom of *B. caeruleus* (Fig 1).

## Ramelteon and Agomelatine strongly promote the recovery of the respiratory function impaired by *B. caeruleus* venom

The most dangerous aspect of the peripheral neuroparalysis caused by kraits, including that caused by *B. caeruleus*, is the respiratory deficit that is at the basis of death following such envenomation [9–20,22,23]. We have devised a simple and sensitive assay of lung ventilation in mice that can be repeated longitudinally on the same animal at different time points after envenomation (Fig 4A). A pressure probe is placed inside the oesophagus at the mediastinum level in anesthetized mice and the recorded signal is characterized by peaks with a frequency corresponding to lung ventilation events. The peak area is related to the volume of air inspired, which is the direct result of the action of the different breathing muscles. The injection in the

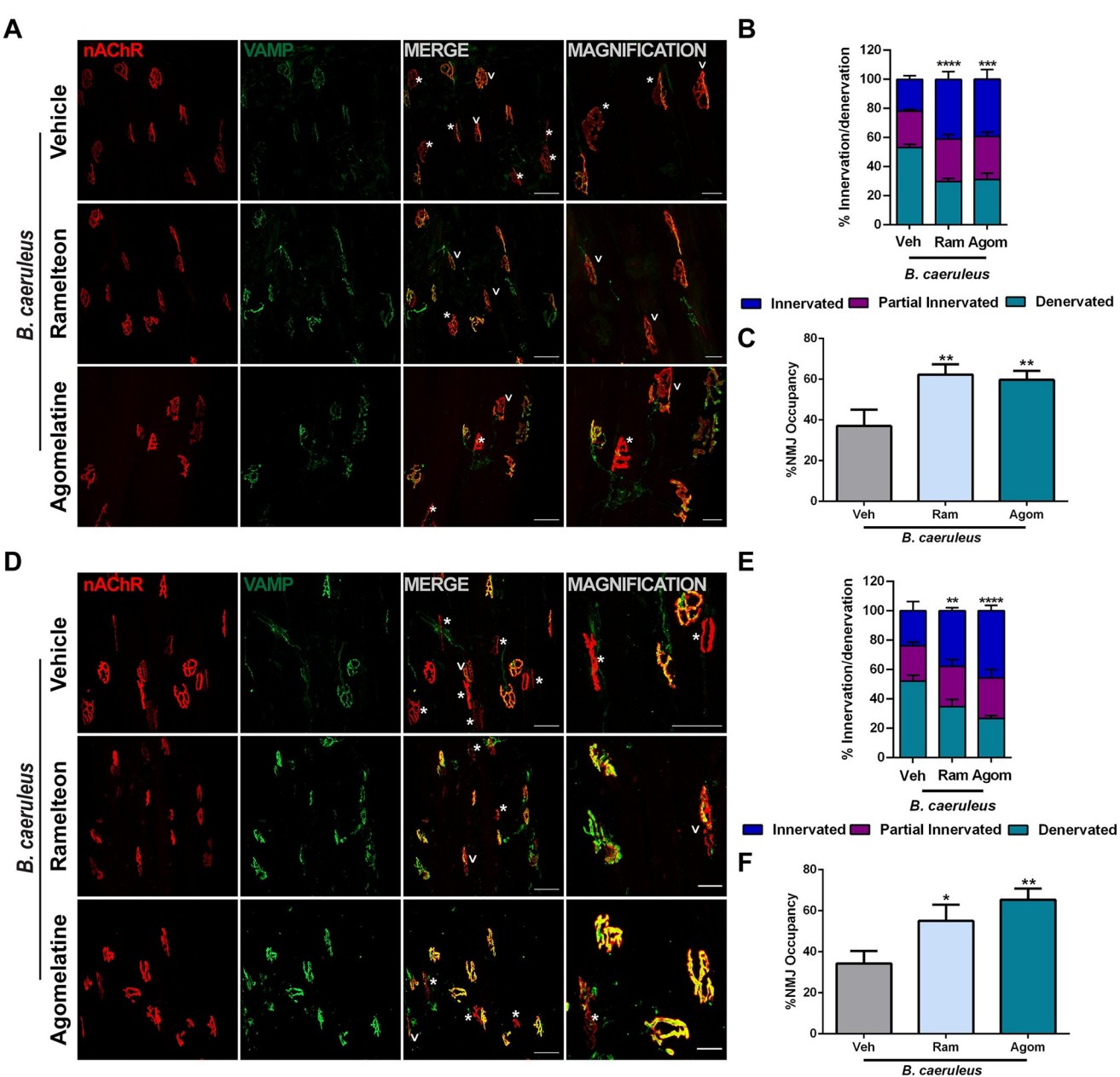

**Fig 3. The MT1 agonists Ramelteon and Agomelatine strongly promote NMJ structural recovery after envenoming with *B. caeruleus*.** Soleus (A) and gastrocnemius (D) muscles were processed for indirect immunofluorescence using fluorescent α-BTX to stain post-synaptic AChRs (*red*), and anti-VAMP1 antibodies to identify the pre-synaptic compartment (*green*). Asterisks identify degenerated NMJs while arrows partial innervated ones. Scale bars: 50 μm (10 μm in magnification). Quantification of degenerated, regenerated and partial innervated NMJs in soleus (B) and gastrocnemius (E) muscles. N = 3, 40 NMJs analyzed/muscle. 2way ANOVA, interaction source of Variation ***p < 0,001, ****p <0,0001 in B, **p < 0,001, ****p <0,0001 in B and E. C and F Percentage of NMJ occupancy, which represents the overlap between pre- and post-synaptic markers. One way ANOVA *p<0,05, **p<0,01.

chest of *B. caeruleus* venom caused a large decrease in the frequency and extent of ventilation at 24 h, with changes in shape and dimension of recorded peaks (Fig 4C). Treatment with Ramelteon or Agomelatine, after venom injection, stimulated the recovery of the respiratory function, as shown in Fig 4B and 4C. Already after 24 h, both drugs caused a large increase in peak area compared to untreated mice, a difference that is more evident four days upon

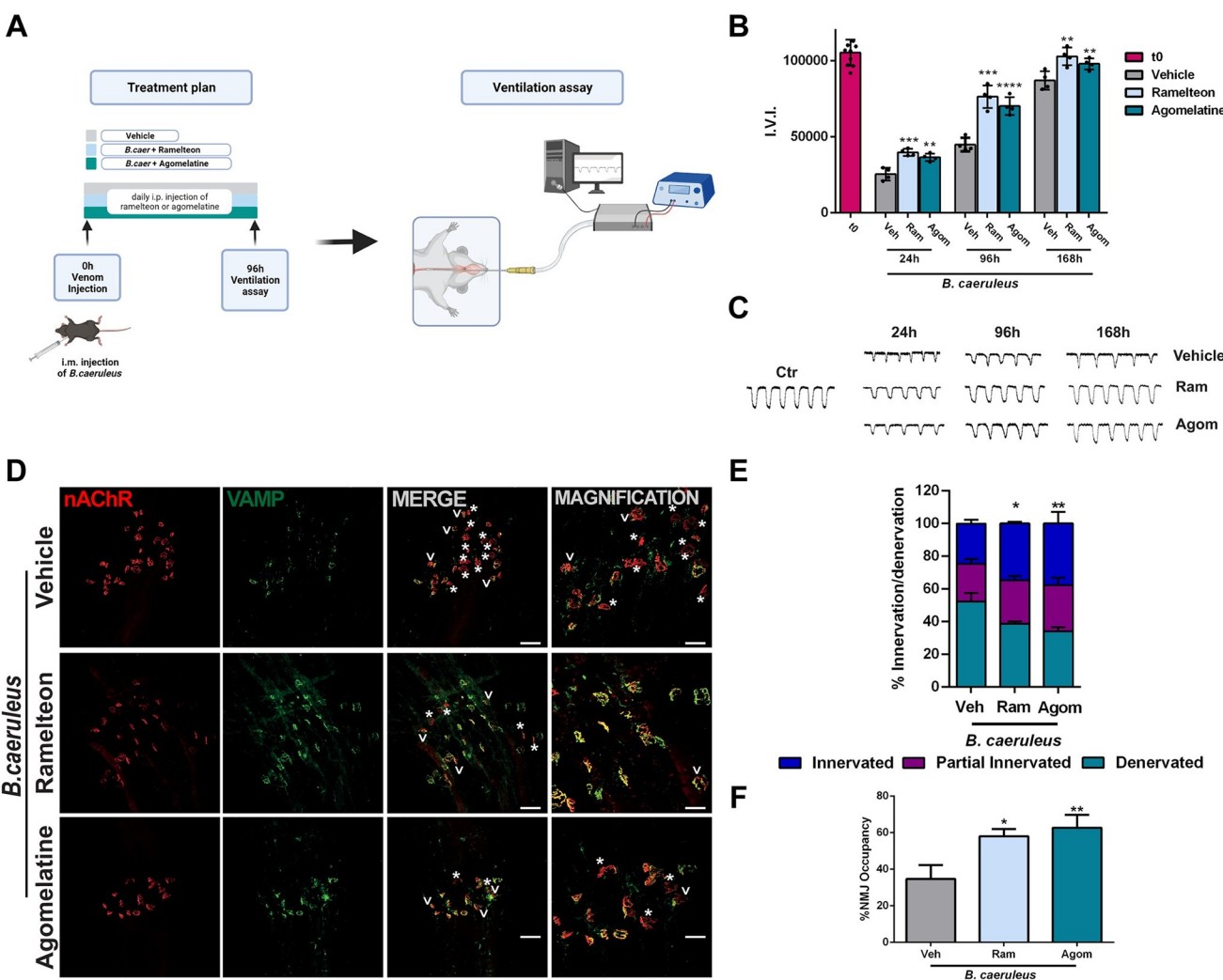

**Fig 4. Recovery of normal respiration after poisoning with *B. caeruleus* venom is strongly stimulated by Ramelteon or Agomelatine.** A) Schematic representation of the technique of measurement employed to calculate the Ventilation Index, Created with BioRender.com. B) Quantitative ventilation assay was performed by measuring the peak area (mV/ms) of consecutive peaks within 20 seconds of trace. The average area was multiplied by the frequency within the time period considered (±SEM). The t = 0 was detected before animals' envenomation, and animals with similar ventilation values were kept to start the experiments. Each group of envenomated animals consisted of $N = 4$ mice. One-way ANOVA **p < 0,001, ****p <0,001 ****p <0,0001. C) Diaphragm muscles were processed for indirect immunofluorescence using fluorescent α-BTX to stain post-synaptic AChRs (red), and anti-VAMP1 antibodies to identify the pre-synaptic compartment (green). Asterisks identify degenerated NMJs while arrows partial innervated ones. Scale bars: 50 μm (10 μm in magnification). D) Quantification of degenerated, regenerated and partial innervated NMJs. N = 3, 40 NMJs analyzed/muscle. 2way ANOVA, interaction source of Variation *p < 0,05, **p <0,01. E) Percentage of NMJ occupancy, which represents the overlap between pre- and post-synaptic elements. One way ANOVA *p < 0,05, **p <0,01.

envenomation. After receiving MT1 receptor agonists mice's lung ventilation did indeed fully recover at 96 hours while control animals did not achieve this outcome until one week later. The results obtained with physical assay of lung ventilation are paralleled by the results of immunofluorescence imaging of the NMJ of pectoral muscles, performed as in Figs 3 and 4, and shown in Fig 2.

## Discussion

Snakebites cause an enormous sanitary problem with a heavy load of physical and psychological sequelae, and a very large number of deaths worldwide. To tackle this global problem, the

World Health Organization has launched a call aimed at improving the therapies to treat snakebite envenomings that include a larger and more focused production of antivenom antisera and the search for small molecules and other therapeutics [1,2,63]. The present work is part of a line of research termed "Innovative research on new therapeutics", dedicated at identifying chemical molecules that could act as therapeutics in the pathology entrained by snakebite envenomings [1,2,64]. This line has already produced promising inhibitors, including Varespladib acting on the presynaptic PLA2 snake neurotoxins [65], NUCC-390 an agonist of the CXCR4 receptor [60,66] and PGDHi, a specific inhibitor of the gerozyme 15-prostaglandin dehydrogenase that promotes the recovery of function of the degenerated NMJ [61].

The main finding of the present work is that the loss of structure and function of motor neuron axon terminals, caused by the venom of *B. caeruleus*, is reversed much more rapidly in mice treated daily for at least 4 days with Ramelteon or with Agomelatine. Considering previous observations on the regenerative properties of NUCC-390 on the peripheral neuroparalysis induced by three different *Bungarus* venoms [66], and by venoms of other neurotoxic snakes including taipan, coral snake and Alpine vipers [9–31,67,68], the therapeutic value of Ramelteon and Agomelatine, in reducing the time of recovery of the respiratory deficit and the NMJ function in general, is likely to be extended to envenoming of many potentially lethal neurotoxic snakes whose mechanism of action is based on the damage to motor nerve terminals.

We have recently documented that the CXCR4 receptor is expressed on the plasma membrane of the damaged neuronal axon and that PSCs, activated by factors released by the damaged axon, produce the CXCR4 ligand, the chemokine CXCL12α which activates CXCR4 thus promoting the regrowth of the axon and the reformation of a functional NMJ. The same growth promoting activity is performed by NUCC-390, a small molecule CXCR4 agonist that targets the motor axon terminal [69,70]. At variance, Ramelteon and Agomelatine act on a receptor, MT1, which is expressed on the surface of PSCs upon motor axon terminal degeneration and they are slightly, but significantly, more potent than NUCC-390, as deduced by the comparison of the present data with those previously reported [59,65–67]. Taken together, the available data emphasize the role of PSCs in the regrowth of the nerve terminal and in the reformation of a functional synapse, and indicate that PSCs are a main target for future investigations aimed at improving nerve terminal regeneration.

A limitation of the use of antisera or PLA2 inhibitors (such as Varespladid) in the clinics, is that they are restricted to a limited time period after bite, because antitoxin antibodies are very effective only until the toxin is exposed and accessible, whilst PLA2 inhibitors are effective only within the few hours taken by the presynaptic snake PLA2 to hydrolyze part of the nerve terminal phospholipids. At variance, an advantage of using drugs acting on receptors (such as NUCC-390, Ramelteon and Agomelatine) or on intracellular enzymes is that they are expected to be effective even after several hours–days from venom injection, though their effectiveness decreases as the length of time elapsed between biting and therapeutic intervention increases. Thus, it is likely that the time window of efficacy of NUCC-390, Ramelteon and Agomelatine is wider than that of antibodies or inhibitors. The time span of effectiveness of the two MT1 agonists employed in the present study remains to be determined, and will be the subject of a following study.

Remarkably Ramelteon and Agomelatine are already used in the clinics for the treatment of insomnia and depression in humans, respectively [49,50], and therefore, are ready for clinical trials. The efficacy of these drugs is not dependent on the particular snake species that caused the neuroparalysis, provided that paralysis is caused by nerve terminal degeneration. Noticeably, these drugs have a long shelf life and do not require refrigeration, two properties particularly relevant in rural tropical settings. In addition, Ramelteon and Agomelatine are not

expensive and are safe even for long treatments [49,50] that exceed the usual recovery from neuro-paralysis. Consequently, it may be possible that higher dose of these drugs can be considered for the use in humans.

## Supporting information

**S1 Fig. Chemical formulas of Melatonin, Ramelteon and Agomelatine.**
(TIF)

**S2 Fig. Structural recovery of respiratory NMJs after poisoning by *B. caeruleus* venom is strongly stimulated by Ramelteon or Agomelatine.** Pectoral (A) and intercostal (D) muscles were processed for indirect immunofluorescence using fluorescent α-BTx to stain post-synaptic AChRs (*red*), and anti-VAMP1 antibodies to identify the pre-synaptic compartment (*green*). Asterisks identify degenerated NMJs while arrows partial innervated ones. Scale bars: 50 μm (10 μm in magnification). Quantification of degenerated, regenerated and partial innervated NMJs in pectoral (B) and intercostal (E) muscles. N = 3, 40 NMJs analyzed/muscle. 2way ANOVA, interaction source of Variation *p < 0,05 in B and ****p <0,0001 in E. C, F) Percentage of NMJ occupancy in pectoral and intercostal muscles, respctively. One way ANOVA **p <0,01, ***p <0,001.
(TIF)

**S1 Data. Excel spreadsheet containing, in separate sheets, the underlying numerical data for Figs 1A, 2B, 2D, 3B, 3C, 3D, 3E, 3F, 4B, 4E, 4F.**
(XLSX)

## Acknowledgments

We thank Dr. W.B. Macklin (Aurora, Colorado) for providing mice expressing the GFP protein in the cytosol of Schwann cells, Prof. M. Solimena for the kind gift of melatonin receptors antibodies. We thank the Referees for their relevant comments.

## Author Contributions

**Conceptualization:** Michela Rigoni, Samuele Negro, Cesare Montecucco.

**Data curation:** Giorgia D'Este, Federico Fabris, Aram Megighian, Samuele Negro.

**Formal analysis:** Giorgia D'Este, Marco Stazi, Aram Megighian, Michela Rigoni.

**Funding acquisition:** Michela Rigoni, Cesare Montecucco.

**Investigation:** Giorgia D'Este, Federico Fabris, Marco Stazi, Chiara Baggio, Morena Simonato, Samuele Negro.

**Methodology:** Giorgia D'Este, Federico Fabris, Chiara Baggio, Morena Simonato, Aram Megighian, Samuele Negro.

**Project administration:** Cesare Montecucco.

**Resources:** Giorgia D'Este, Federico Fabris, Morena Simonato, Samuele Negro.

**Supervision:** Cesare Montecucco.

**Writing – original draft:** Samuele Negro, Cesare Montecucco.

**Writing – review & editing:** Samuele Negro, Cesare Montecucco.

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
