## [Decision Letter · Decision Letter 0]

14 Dec 2023

Dear Dr Negro,

Thank you very much for submitting your manuscript "Agonists of melatonin receptors strongly promote the functional recovery from the neuroparalysis induced by neurotoxic snakes" for consideration at PLOS Neglected Tropical Diseases. As with all papers reviewed by the journal, your manuscript was reviewed by members of the editorial board and by several independent reviewers. The reviewers appreciated the attention to an important topic. Based on the reviews, we are likely to accept this manuscript for publication, providing that you modify the manuscript according to the review recommendations. 

Sincerely,

José María Gutiérrez

Section Editor

Section Editor: The reviewers gave a positive evaluation of this study, while two of them raised a few methodological concerns that need to be carefully addressed when preparing a revised version of the manuscript

Reviewer's Responses to Questions

**Key Review Criteria Required for Acceptance?**

**Methods**

-Are the objectives of the study clearly articulated with a clear testable hypothesis stated?

-Is the study design appropriate to address the stated objectives?

-Is the population clearly described and appropriate for the hypothesis being tested?

-Is the sample size sufficient to ensure adequate power to address the hypothesis being tested?

-Were correct statistical analysis used to support conclusions?

-Are there concerns about ethical or regulatory requirements being met?

Reviewer #1: The objectives of the study are clearly articulated with a clear testable hypothesis stated.

The study is appropriately designed to address the stated objectives.

The population is clearly described and appropriate for the hypothesis being tested.

The sample size is sufficient to ensure adequate power to address the hypothesis being tested.

The correct statistical analysis was used to support conclusions.

There are no concerns about ethical or regulatory requirements.

Reviewer #2: see reviewer's report

Reviewer #3: Evoked Junctional Potentials (EJPs) recordings and Compound muscle action potential (CMAP) recordings

How long did it take to inject ramelteon and agomelatine after the snake venom? Were mice maintained anesthetized for 96 h? Explain in detail the general procedure.

**Results**

-Does the analysis presented match the analysis plan?

-Are the results clearly and completely presented?

-Are the figures (Tables, Images) of sufficient quality for clarity?

Reviewer #1: The analysis presented match the analysis plan.

The results are clearly and completely presented.

The figures (Tables, Images) are of sufficient quality for clarity.

Reviewer #2: YES, YES, YES

Reviewer #3: (No Response)

**Conclusions**

-Are the conclusions supported by the data presented?

-Are the limitations of analysis clearly described?

-Do the authors discuss how these data can be helpful to advance our understanding of the topic under study?

-Is public health relevance addressed?

Reviewer #1: The conclusions are supported by the data presented.

The limitations of analysis are clearly described.

The authors discuss how their data can be helpful to advance our understanding of the topic under study.

The public health relevance is clearly addressed.

Reviewer #2: Conclusions are supported by the data presented. The authors comment on possible practical application of the study.

Reviewer #3: Have you tried to mimic human envenomation and use ramelteon and/or agomelatine after 3-24 after snake venom injection? Do they accelerate the outcome if they are associated with a specific antivenom?

**Editorial and Data Presentation Modifications?**

Reviewer #1: The work is meticulously performed and the manuscript clearly written. The results presented in this paper are medically very important. 

No modifications are needed. In my opinion, this work should be accepted for publication without a delay.

Reviewer #2: (No Response)

Reviewer #3: (No Response)

**Summary and General Comments**

Reviewer #1: In this paper, the authors (PNTD-D-23-01512) presented the effects of two drugs, Ramelteon and Agomelatine, licensed for the treatment of insomnia and depression, both agonists of melatonin receptor 1 (MT1), which plays a major role in the recovery of function of the neuromuscular junction (NMJ) after the degeneration of motor axon terminals due to the action of snake venom neurotoxic sPLA2s. The key finding of this work is that the loss of structure and function of the motor neuron axon terminals, after the envenomation of the mouse by the neurotoxic snake (krait) venom, is reversed much more rapidly if treated for at least 4 days with Ramelteon or Agomelatine. The authors propose that these drugs, which are commercially available, safe, non-expensive, have a long bench life and can be administered long after a snakebite even in places far away from health facilities, should be tested on human patients bitten by neurotoxic snakes acting presynaptically for preventing the development of a deadly respiratory paralysis and to speed up their recovery. 

The work is meticulously performed and the manuscript clearly written. The results presented in this paper are medically very important. 

No modifications are needed. In my opinion, this work should be accepted for publication without a delay.

Reviewer #2: In previous work, Monteccuco´s group showed that the intercellular signaling axis melatonin-melatonin receptor 1 (MT1) plays a major role in the recovery of function of the NMJs after degeneration of motor axon terminals caused by krait envenomings. Now the authors report on the very promising effect of Ramelteon and Agomelatine, two agonists of MT1 on sell in pharmacies for treatment of insomnia and depression, on the recovery of NMJ function after degeneration of motor axon terminals caused by the venom of Bungarus caeruleus. These drugs cause a relevant recovery of the respiratory function with respect to mice treated with vehicle already after 24 hours from venom injection. Based on these results, the authors suggest that Ramelteon and Agomelatine should be tested for their potential action on the recovery of normal physiology in neuroparalysed patients bitten by neurotoxic snakes. 

This is a very interesting investigation with high potential of application in the treatment of envenoming by kraits and other neurotoxic snakes whose venom's mechanism of action is based on nerve terminal damage. I have just a minor comment concerning the dosing of venom and the MT1 agonists:

Mice (25-30 g) "were locally injected in the hind limb with 36 μg/Kg of B. caeruleus venom in 15 μl PBS containing 0.2% gelatin." Reported LD50 for B. caeruleus is 0.1-0.3 (i.v.)/0.45 (i.m.) μg/g mouse body weight. 36 μg/Kg of B. caeruleus venom of venom equals 0.036 μg/g mouse body weight or 0.08-0.36 LD50s. Right? Why this low dosing? On the other hand, "Mice were daily locally injected in the soleus muscle with Ramelteon (29 μg/Kg), diluted in 20 μl PBS containing 0.2% gelatin) or Agomelatine (0.36 mg/Kg). These doses equal 0.8 μg Ramelteon/ μg B. caeruleus venom, 10 μg Agomelatine/ug B. caeruleus. "For Ramelteon we used a dose that corresponds to one fourth of that suggested for long-term use in humans (8 mg/day, taking 70 Kg as average weight), while for Agomelatin we employed the same dose used in humans (25 mg/day)". Dose translation between animal models and human is usually done applying allometric dose scaling (i.e., Reagan-Shaw, S., Nihal, M., Ahmad, N. (2007). Dose translation from animal to human studies revisited. FASEB J. 22, 659–661; Sharma and McNeill, British Journal of Pharmacology (2009) 157, 907–921; doi:10.1111/j.1476-5381.2009.00267; dosing in mice (mg/Kg) = Dosing in human (mg/Kg) x (Km human/Km mouse). One fourth of 8 mg/70 Kg = 29 μg/Kg (= 0.73-0.87 mg/ 25-30g mouse), whereas allometric translation would suggest mouse (μg/Kg)= 29 μg/Kg * (37/3) = 357.6 μg/Kg = 8.9-10.7 mg (25-30g mouse). Similarly, human dose of 0.36 mg/Kg would correspond to 360 * (37/3) = 4440 μg/Kg = 111-133 mg (25-30g mouse)... Please, comment why linear rather than allometric dose scaling was chosen...

Reviewer #3: This is a very interesting manuscript reporting the use of ramelteon and agomelatine to combat the paralysis evoked by Bungarus caeruleus. Generally, the paper is well written, but I have a few comments regarding the Methods and how ramelteon and agomelatine were used.

PLOS authors have the option to publish the peer review history of their article (what does this mean?). If published, this will include your full peer review and any attached files.

Reviewer #1: No

Reviewer #2: Yes: Juan J Calvete

Reviewer #3: No

Figure Files:

Data Requirements:

Reproducibility:

References

---

## [Editor Report · Decision Letter 1]

22 Dec 2023

Dear Dr Negro,

We are pleased to inform you that your manuscript 'Agonists of melatonin receptors strongly promote the functional recovery from the neuroparalysis induced by neurotoxic snakes' has been provisionally accepted for publication in PLOS Neglected Tropical Diseases.

Best regards,

José María Gutiérrez

Section Editor

The comments of the reviewers have been adequately addressed in the revised version of this manuscript.

---

## [Editor Report · Acceptance letter]

1 Jan 2024

Dear Dr Negro,

We are delighted to inform you that your manuscript, "Agonists of melatonin receptors strongly promote the functional recovery from the neuroparalysis induced by neurotoxic snakes," has been formally accepted for publication in PLOS Neglected Tropical Diseases.

Best regards,

Shaden Kamhawi

co-Editor-in-Chief

Paul Brindley

co-Editor-in-Chief
